

# Hydroxyapatite/calcium alginate composite particles for hemostasis and alveolar bone regeneration in tooth extraction wounds

Gang He[1],[*], Zhihui Chen[2],[*], Luyuan Chen[2], Huajun Lin[2],
Chengcheng Yu[2], Tingting Zhao[1], Zhengwen Luo[2], Yuan Zhou[1],
Siyang Chen[2], Tianjiao Yang[2], Guixian He[2], Wen Sui[3], Yonglong Hong[2]
and Jianjiang Zhao[1]

[1] Shenzhen Stomatological Hospital, Southern Medical University, Shenzhen, China
[2] Department of Maxillofacial Surgery, Shenzhen Hospital, Southern Medical University, Shenzhen, China
[3] College of Pharmacy, Shenzhen Technology University, Shenzhen, China
[*] These authors contributed equally to this work.

Corresponding authors
Yonglong Hong, ylhong93@163.com
Jianjiang Zhao, zjj2521@sina.com

## ABSTRACT

Tooth extractions can lead to complications such as post-extraction bleeding and bone resorption, which may result in unfavorable outcomes for implant restoration afterwards. To strive for an optimal condition for further restoration procedures, appropriate strategies, such as hemostasis or bone regeneration, are encouraged to be employed. However, the existing products are failed to meet both needs. As a widely employed tissue engineering materials, hydroxyapatite and calcium alginate both have demonstrated excellent performance in osteogenesis. However, their inferior mechanical strength poses a major limitation to their use in supporting the contracted extraction socket, which can easily lead to alveolar crest atrophy and barely achieve satisfying results. Calcium alginate improves the mechanical strength of hydroxyapatite, enabling the formation of new bone tissue and degradable *in vivo*. In this study, we demonstrated the biocompatibility and haemostatic ability of nCA particles on a rat tooth extraction model. In addition, long-term observation has revealed minimal inflammation and bone tissue regeneration. Our findings suggest a promising insight for clinical applications in hemostasis and bone regeneration after tooth extraction.

## INTRODUCTION

When post-extraction bleeding persists for more than 8 to 12 h after the dental extraction procedure, improper management may result in life-threatening complications (*Ono et al., 2023*). After an extraction, surgical hemostatic agents are commonly used, but their efficacy may be limited, particularly in patients receiving long-term anticoagulation therapy (*Radhakrishna, Shukla & Shetty, 2023*). Before performing extractions, dentists

and oral surgeons are required to evaluate the risk of bleeding. Patients on anticoagulants or with bleeding disorders may require special care and management to prevent excessive bleeding (*Costache et al., 2023*). In addition, diligent post-operative care and monitoring are essential for the prompt detection and treatment of any surgical complications. Various hemostatic agents can be used to stop bleeding after tooth extraction (*Jiao et al., 2023*). Local measures may include compression, sutures, and topical agents oxidized cellulose (*Abdullah & Khalil, 2014*) and collagen (*Baumann et al., 2009*). Under specific conditions, systemic hemostatic agents, such as tranexamic acid (*Carter et al., 2003*), may be necessary to achieve adequate hemostasis. However, the risk of thromboembolic complications must be carefully weighed against the use of the aforementioned agents in patients receiving anticoagulation therapy.

Alveolar bone resorption, which is frequently encountered in clinical practice, can impede the rehabilitation process following tooth extraction. To address this issue, techniques for preserving the alveolar ridge have been developed (*Al-Badran, Bierbaum & Wolf-Brandstetter, 2023*). By preventing alveolar ridge resorption and promoting bone formation, preservation techniques, such as guided bone regeneration, promote bone formation. In order to fill bone defects in the oral cavity, both soft and hard tissues, and not just the supportive properties of the implanted material, should be considered (*Wu et al., 2023*). Autografts, allografts, xenografts, and synthetic biomaterials are among the bone graft materials that can be utilized for alveolar ridge preservation. However, conventional bone scaffold biomaterials may have certain limitations, such as donor site trauma and inflammation (*Pan, Ye & Wu, 2021*). Due to the fact that each material has its own advantages and disadvantages, judicious consideration must be given to the most suitable technique and material for each specific situation.

Due to its similarity to natural bone tissue, hydroxyapatite (HAp) is widely utilized as an exceptional biomaterial for bone tissue engineering (*Darjanki et al., 2023*). As a result, HAp has been extensively studied as a bone substitute material, especially in plastic surgery, dentistry, and maxillofacial surgery. Even though HAp scaffolds have excellent biocompatibility and osteoconductive properties, there are limitations in terms of grafting and fixing bone defects (*He et al., 2023*). Consequently, researchers are developing novel strategies to enhance the mechanical and biological properties of HAp scaffolds. Combining HAp composites with other biomaterials, such as ceramics or polymers, could potentially circumvent these limitations. These composites exhibit a significant increase in mechanical strength, biodegradability, and osteoinductivity. For instance, HAP/poly (lactic-co-glycolic acid) (PLGA) composite scaffolds demonstrate enhanced properties and bone regeneration (*Liu et al., 2023*). However, the relatively high cost of PLGA material places an economic burden on patients and restricts its widespread clinical application (*Jem & Tan, 2020*). Another method for enhancing HAP properties is to modify the scaffold's surface for better integration with the surrounding tissue. There have been numerous approaches to surface modification, including coating scaffolds with extracellular matrix proteins or growth factors that promote cell adhesion, proliferation, and differentiation (*Sindhya et al., 2023*). These modifications significantly enhance the bioactivity of the scaffolds, resulting in enhanced integration with the surrounding tissues.

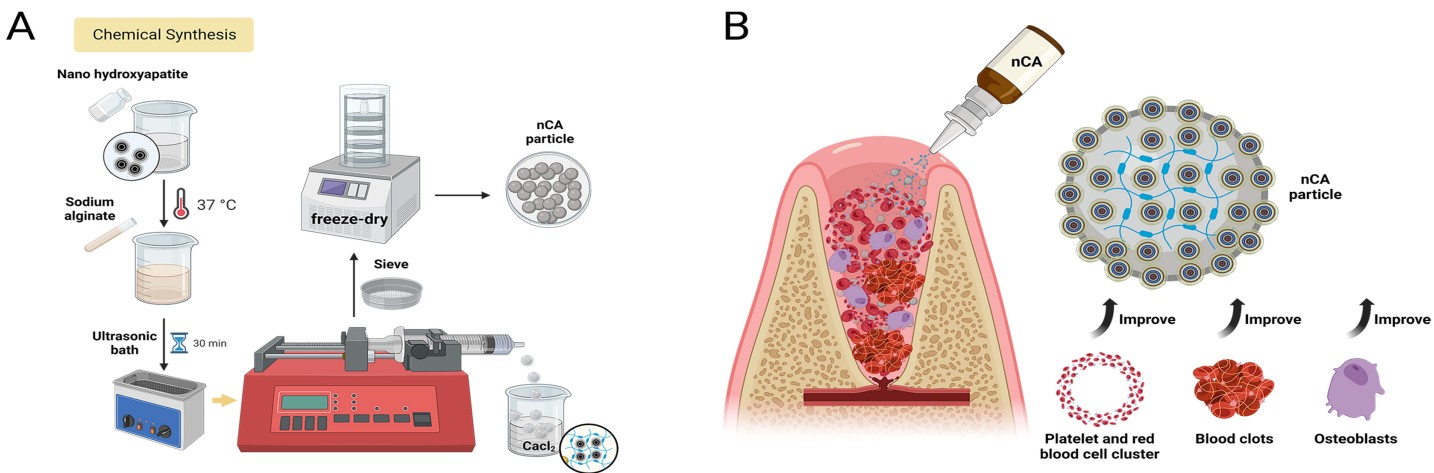

**Scheme 1 Schematic illustration for the preparation and application of particles.** (A) Chemical structure of the nCA particles and (B) schematic illustration depicting the mechanism of particles used in extraction wounds to prevent hemostasis and promote bone regeneration. Created in BioRender.com

Nonetheless, surface modification of HAp can also have unfavorable effects, such as a decrease in mechanical strength and an accelerating degradation rate of HAp. This further reduces its lifespan. In addition, surface-modified HAp may not only increase the financial burden on patients to varying degrees, but may also prevent precise control over the amount and shape of scaffolds, resulting in poor reproducibility and uncertain outcomes. In order for HAp to fulfill its potential as a bone regeneration material, it is crucial to investigate new techniques for enhancing its properties.

Calcium alginate is incorporated into composite scaffolds to circumvent the limitations of sole HAp (*Wang et al., 2023*). Due to its natural origin and distinctive monomer block structure, alginate is an excellent material for hydrogel development (*Paques et al., 2014*). For the preparation of alginate hydrogels, numerous techniques have been utilized, including the ionic cross-linking reaction with divalent cations ($Ca^{2+}$, $Mg^{2+}$, and $Fe^{2+}$). Calcium chloride ($CaCl_2$) is commonly used to produce calcium alginate hydrogels by utilizing $Ca^{2+}$ for cross-linking. Calcium alginate hydrogels have a distinctive three-dimensional network structure (*Zou et al., 2020*), which enables the effective sequestration and sustained release of drugs or proteins at the target sites. In summary, among the various biomaterials used, calcium alginate hydrogels have the greatest potential for tissue engineering, particularly in bone regeneration.

The objective of this study is to create an *in situ* hydroxyapatite/calcium alginate composite infill particle (Scheme 1) that is biocompatible, biodegradable, and easy to access. It is hypothesized that the synergistic effect of two kinds of material will increase the adhesion and aggregation of red blood cells. This will allow for the formation of a solid blood clot, thereby enhancing hemostatic properties and bone regeneration. Therefore, we systematically evaluated the particle size, surface morphology, and swelling ratio of the composite material's physicochemical properties. This was followed by the *in vivo* and *in vitro* investigations of biocompatibility, osteogenic capacity, and biodegradability. Finally, the tooth extraction model of a rat was used to examine the hemostatic effect and bone

regeneration capacity of the granular material. Our research may, thus, shed light on post-operative therapy and tissue regeneration with upcoming agents.

## MATERIALS AND METHODS

### Materials

Nanohydroxyapatite (nHAp) was obtained from Boer Co., Ltd. (Shanghai, China; 20 nm). Calcium chloride and sodium alginate (SA) were purchased from Aladdin Co., Ltd. All other chemicals were purchased from commercial sources and had an analytical purity of 99.9% or higher.

### Synthesis of nCA

A 5% nHAp solution was prepared with deionized water. Immediately after weighing the SA, it was added to the nHAp solution, resulting in a 2% nHAp/SA solution. The solution was shaken overnight at 37 °C and then subjected to ultrasonic treatment to eliminate any air bubbles. Subsequently, using a syringe pump, syringes filled with 5 mL of solution were used to slowly inject the solution into 100 mL of 1% calcium chloride during an ion-exchange cross-linking reaction. The final nCA particles were obtained by freeze-drying the composite hydrogel for 24 h. In addition, pure CA was synthesized in the same manner without the addition of nHAp.

### Scanning electrical microscopy

The surface morphology of the as-obtained particles was observed using SEM (JSM-6700F; JEOL, Tokyo, Japan).

### Swelling ratio test

The swelling behavior of CA ($n = 6$) and nCA ($n = 6$) was determined by submerging them in PBS at 37 °C. We measured the initial weight ($W_0$) and the weight of the particles at regular intervals ($W_t$) until equilibrium was achieved. The following equation was then used to calculate the swelling rates:

$$Swelling\ Ratio\ (\%) \ = \ (W_t - W_0)/W_0 \times 100\%. \tag{1}$$

### Degradation behaviors

The initial weight ($W_0$) of CA ($n = 6$) or nCA ($n = 6$) was recorded. Using a constant-temperature shaker, the particles were then immersed in 30 mL of PBS and then subjected to a degradation test at 37 °C. Subsequently, we measured the dry weight ($W_t$) of the particles after they were removed from the PBS solution, rinsed, and lyophilized. To determine the degradation of materials, the following equation was used:

$$Mass\ (\%) \ = \ W_t/W_0 \times 100\%. \tag{2}$$

### *In vitro* biocompatibility

In this study, a cell counting kit-8 (CCK-8) (Dojindo, Kumamoto, Japan) ($n = 6$) was used for cell counting. We seeded bone marrow mesenchymal stem cells (BMSCs) (Topbiotech,

China) in 96-well plates at a density of $2 \times 10^3$ cells per well. After 24 h of incubation, a fresh medium containing varying concentrations of nCA (2 mg/mL, 3 mg/mL, 4 mg/mL, and 5 mg/mL) was substituted for the nCA groups. Using an enzyme marker at 450 nm (Tecan, Switzerland), the cell viability was measured after 1, 3, and 5 days of incubation with the CCK-8 solution. Consequently, the OD (450 nm) values for the experimental group, the NC group, and the background group were recorded as $OD_E$, $OD_{NC}$, and $OD_B$, respectively. The formula below was used to calculate cell viability (%):

$$\text{Cell Viability (\%)} = (OD_E - OD_B)/(OD_{NC} - OD_B) \times 100\%. \tag{3}$$

Staining for live or dead cells was performed using the Live/Dead Viability/Cytotoxicity Kit (Invitrogen). Accordingly, the BMSCs were incubated in co-culture medium containing 5 mg/mL CA or 5 mg/mL nCA. Green and red fluorescences were then visualized using a fluorescence microscope (Leica Microsystems CMS GmbH, Germany) after 1, 3, and 5 days of incubation.

## Hemocompatibility tests

The hemolysis ratio was used to evaluate the hemocompatibility of particles. The red blood cells (RBCs) were separated by centrifuging a sample of fresh citrate-anticoagulated rat whole blood at 3,000 rpm for 10 min. The RBCs were then washed and centrifuged three times in PBS, yielding a suspension of RBCs (4 vol.%) in PBS. During incubation, the PBS suspension and the different particles were mixed together with the PBS suspension. Subsequently, the mixtures are centrifuged at 3,000 rpm for 10 min, at which point the supernatants are ready for analysis (duplicated samples, $n = 6$). A microplate reader (Tecan, Switzerland) was then used to measure the absorbance (Abs) at 540 nm of each supernatant. Triton X-100 (0.1%) solution and a PBS solution were used as the positive and negative control groups, respectively. The hemolysis ratio (%) was then calculated as follows:

$$\text{Hemolysis ratio (\%)} = (\text{Abs}_{\text{sample}} - \text{Abs}_{(-)})/(\text{Abs}_{(+)} - \text{Abs}_{(-)}) \times 100\% \tag{4}$$

## Alkaline phosphatase (ALP) staining

After 7 days of osteogenic induction, BMSCs were fixed with 4% paraformaldehyde (PFA) at room temperature for 15 min, followed by three washes with PBS. The BCIP/NBT Kit (CWBIO, Beijing, China) was applied according to the manufacturer's instructions. BMSCs were stained for 30 min in the dark with BCIP/NBT dyes. The cells were then washed three times with PBS and observed under a microscope.

## Alizarin Red S (ARS) staining

Following 14 days of osteogenic induction, BMSCs were fixed and rinsed according to the protocol for ARS staining. Subsequently, the ARS (Sigma-Aldrich) solution was applied to the cells for 15 min at room temperature, followed by three PBS rinses. After the stained cells were air-dried, images were captured.

 

### *In vivo* rat extraction and implant materials

Before the experiment began, Sprague-Dawley (SD) rats (8 weeks, 250–300 g; Vital River Laboratory Animal Technology Co., Ltd., Beijing, China) were acclimatized for a week. A standard 12-h light-dark cycle was maintained, and rats were fed rat chow (Sniff, Soest, The Netherlands) and water *ad libitum* under standard laboratory conditions. All animal experiments were conducted in accordance with protocols approved by the ethical committee of the Southern University of Science and Technology (Protocol No. SUSTech-JY202109009). The animals were anesthetized by inhaling isoflurane and euthanized by inhaling $CO_2$.

The effect of nCA particles on alveolar bone regeneration was assessed using a classic rat alveolar bone regeneration model. The left mandibular incisors of rats were drilled every 3 days. After three consecutive cuts, the left incisors were removed under intraperitoneal injection of 2% pentobarbital sodium (50 mg/kg). The rats were then randomly divided into three groups: the CA group ($n = 6$), the nCA group ($n = 6$), and the control group ($n = 6$). In the CA and nCA groups, the extraction sockets were filled with the particles without sutures. In the control group, cotton rolls were used to compress the extraction sockets without any filling material. All rats received intraoperative and post-operative penicillin injections (80,000 U, intramuscular).Subsequently, we euthanized half of all rats after 1 and 4 weeks, and then fixed their mandibular specimens for 48 h in 4% paraformaldehyde. The fixed specimens were then examined histologically and scanned with a micro-CT after immersion in PBS.

### Evaluation of hemostatic efficiency

For each group, bleeding duration and weight were determined. During bleeding time, extraction wound hemostasis is completed (*Zhu et al., 2022*). The bleeding weight is the difference in weight between the cotton roll prior to and following aspiration (*Bu et al., 2019*). After achieving hemostasis, the animals were observed for 3 min to ensure that hemostasis had been achieved.

### *In vivo* biocompatibility

In order to evaluate *in vivo* biocompatibility, CA/nCA particles were subcutaneously implanted into the backs of rats. After 7 days, the rats were sacrificed and the particles and surrounding tissues were removed. After fixation in paraform, dehydration, embedding, and sectioning, tissue samples were embedded in 5 $\mu$m-thick sections. Sections stained with hematoxylin and eosin (H&E) were examined using a pathology section scanner (NanoZoomer S60; Hamamatsu, Shizuoka, Japan).

### Micro-CT evaluation

Micro-computed tomography (Micro-CT) was used to evaluate the formation of new bone in the rat mandibular defect area. Due to the low bone formation at 7 days, our analysis of the CT data was limited to 28 days. In order to evaluate the quality and quantity of new bone tissue, analyses were performed on the trabecular number (Tb.N), trabecular thickness (Tb.Th), and trabecular bone separation (Tb.Sp) of the target area. CTAn

software was utilized to evaluate the extent and proportion of new bone formation. CTvox software was utilized to reconstruct mandibular alveolar bone following extraction in order to assess alveolar ridge preservation. The new bone area was determined by dividing the new bone area by the alveolar area in sagittal and coronal CT sections corresponding to three-dimensional (3D) reconstruction images. Using the methods, the visual angle of images was determined in order to maintain measurement criteria consistency.

All measurements were conducted independently by two analysts, and the results were analyzed using statistical software.

## Histological analysis

Following decalcification, the fragments of the mandible were prepared for histological examination. The fragments were embedded in paraffin, sliced into thin sections, and stained with hematoxylin and eosin (H&E). The sections were examined using a pathology slide scanner (NanoZoomer S60; Hamamatsu, Shizuoka, Japan). To ensure accuracy and consistency, the histological evaluation was conducted by two independent inspectors. The images were evaluated for the quantity and quality of new bone formation, as well as for indications of inflammation, necrosis, and other abnormalities.

## Statistical analysis

All information was presented as the mean $\pm$ standard deviation (s.d.). In this study, GraphPad Prism 9 was used for statistical analysis. For statistical analysis, unpaired Student's t-tests and one-way analysis of variance (ANOVA) were employed. The level of statistical significance was set at $p = 0.05$ ($^*p < 0.05$, $^{**}p < 0.01$, $^{***}p < 0.001$, $^{****}p < 0.001$ and not significant (*ns*) with $p > 0.05$).

# RESULTS AND DISCUSSION

## Preparation and characterization of nCA

The production of gel spheres relied on microfluidic technology based on droplets. The cross-linking reaction of calcium ions present in the calcium chloride solution and the formation of hydrogel spheres generated microdroplets of the mixture (*Li et al., 2023*). By adjusting the flow rate of the mixture and the size of the nozzle, the size and shape of the spheres can be controlled (*Bi, Lin & Deng, 2019*). Correspondingly, these hydrogel spheres can serve as scaffolds for tissue regeneration and drug delivery, and their physical properties can be tailored for particular applications. In addition, its ease of preservation and administration makes it a promising candidate for applications in wound healing and tissue engineering. The white color of nHAp particles rendered the nCA hydrogel spheres opaque white (Figs. 1B, 1D and 1F). In contrast, the transparency of the CA hydrogel resulted from the absence of nHAp particles (Figs. 1A, 1C and 1E). The incorporation of nHAp into CA hydrogel not only strengthened the combined particles but also improved their morphology. However, owing to the low strength of CA, pure CA produced irregularly shaped particles (Figs. 1G and 1I), whereas the addition of nHAp resulted in a more spherical shape of the nCA particles (Fig. 1H). This was crucial for the reproducibility of the morphology and functionality of the particles.

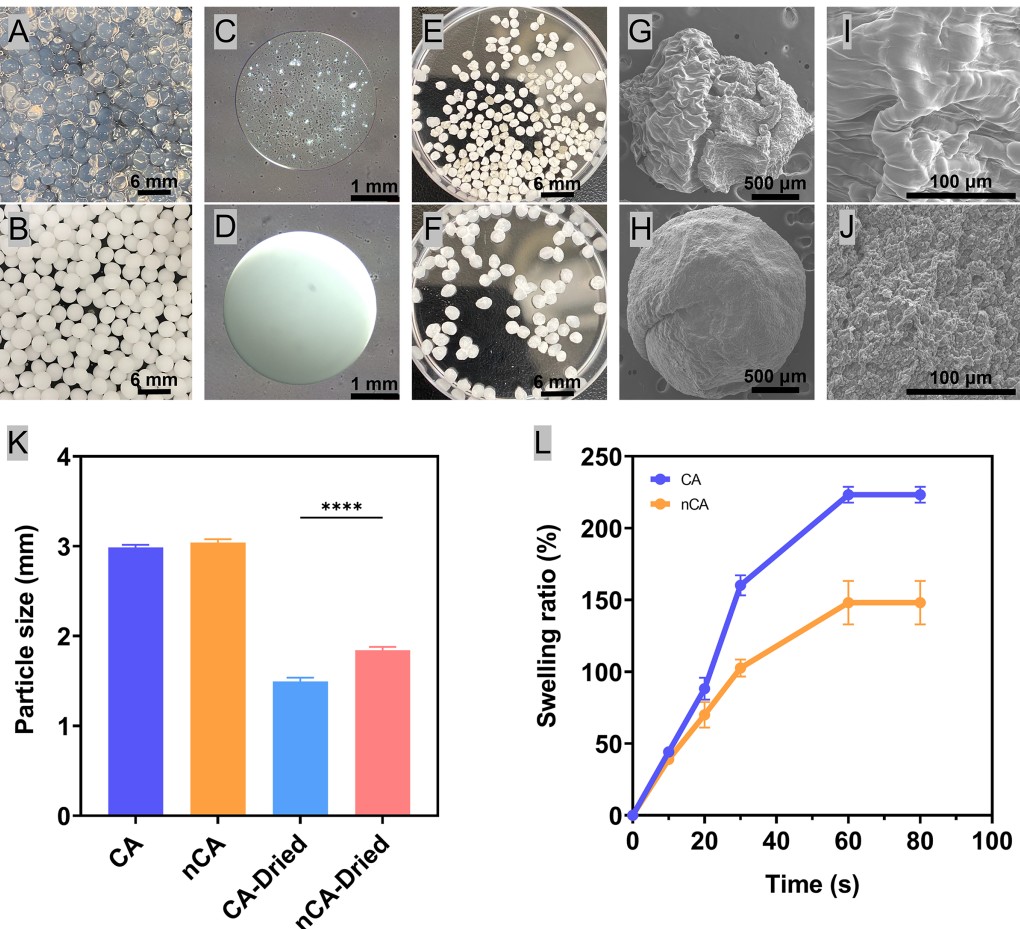

**Figure 1 Preparation and characterization of nCA particles.** (A) CA hydrogel. (B) nCA hydrogel. (C) Micrograph of a CA hydrogel sphere. (D) Micrograph of an nCA hydrogel sphere. (E) CA particles. (F) nCA particles. (G) SEM images of CA. (H) SEM images of nCA. (I) SEM images of CA with local details. (J) SEM images of nCA with local details. (K) Particle size of CA and nCA under freeze-dried and non-freeze-dried conditions (****$p < 0.0001$). (L) Swelling ratios of freeze-dried particles.

The presence of nHAp particles on the surface of nCA (Fig. 1J) is likely due to the addition of nHAp to the CA solution prior to gelation. During gelation, nHAp particles were captured in the CA hydrogel matrix, resulting in the observed morphology of nCA particles. These nHAp particles may also provide an additional contact surface for interactions between biomolecules and cells.

In addition, the lyophilization process yielded a product with stable properties that could be stored and transported with ease. Correspondingly, the hydrogel spheres shrank after freeze-drying due to dehydration, and the particle size of the freeze-dried nCA spheres was greater than that of CA due to the presence of hydroxyapatite (Fig. 1K). Moreover, the addition of nHAp made the particle network denser and less susceptible to expansion. Materials used for extraction hemostasis must have an appropriate swelling rate to prevent excessive pressure on the socket, which can lead to bone resorption (*Zhao et al., 2023*). The lyophilized granular material absorbed water instantly and reached equilibrium

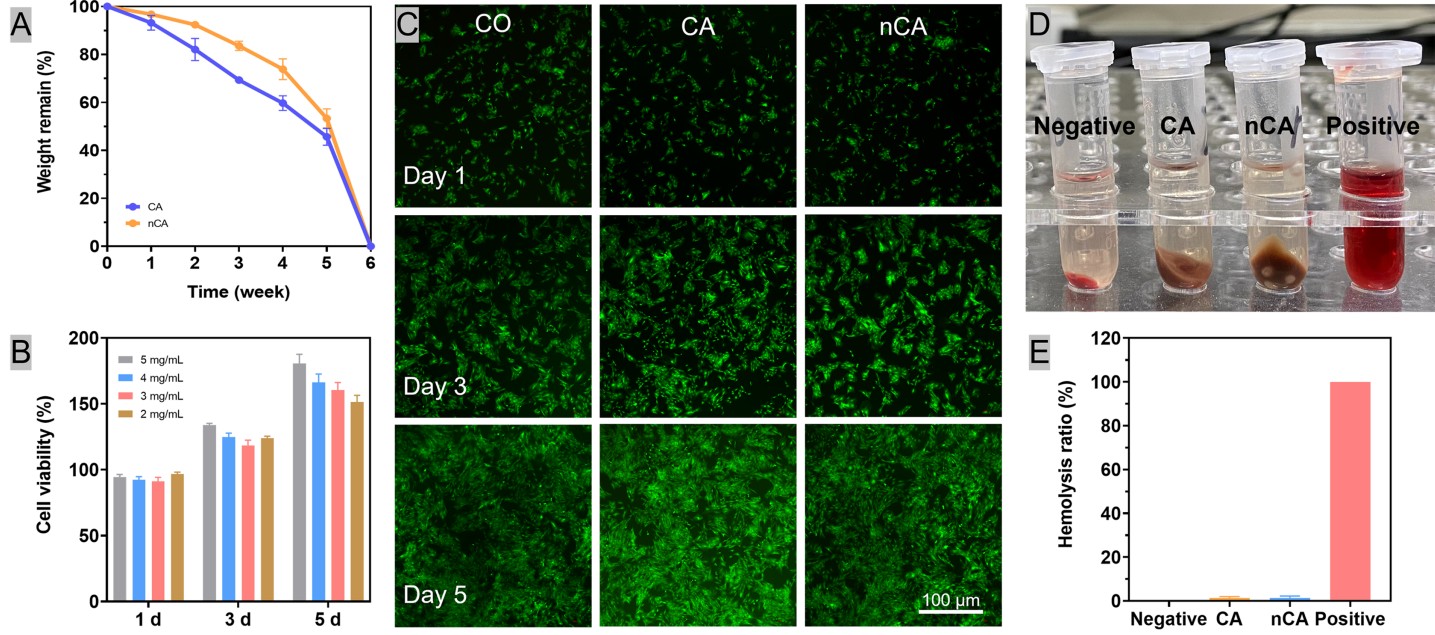

**Figure 2 Biocompatibility of nCA *in vitro*.** (A) *In vitro* degradation curves of CA and nCA in PBS. (B) Cell viability of BMSCs after incubation with nCA for 1, 3, and 5 days. (C) The representative images of BMSCs with Dead/Live staining (green: live; red: dead) after incubation with particles (5 mgmL$^{-1}$) for different periods. (D, E) Hemolytic ratios of all particles.

in approximately 1 min. The addition of nHAp kept the granular network more compact and less susceptible to excessive swelling, which decreased the swelling rate from more than 220% of CA to less than 150% of nCA (Fig. 1L). It is essential for the material to have a certain degree of swelling in order to maintain the shape of the alveolar bone without the formation of cavities within the wound, which could lead to infection (*Sukegawa et al., 2019*). There is a growing interest in the development of composite materials for tissue engineering applications (*Alam et al., 2023*). Composites exhibit enhanced mechanical, biological, and functional properties as a result of the combination of diverse materials (*Collins et al., 2021*) Encapsulating hydroxyapatite in calcium alginate can facilitate the controlled release of bioactive molecules to promote bone restoration and regeneration. In addition, the composite scaffolds can offer a customized microenvironment for cells to simulate their proliferation, differentiation, and matrix production. The combination of alternative materials can result in improved properties and functions, allowing for more efficient restoration and regeneration of damaged tissues (*Wu et al., 2017*).

### *In vitro* degradation

To achieve optimal results, the rate of degradation of the granular material should correspond with the rate of bone tissue healing. The addition of nHAp to nCA gel strengthened the network, delayed the release of nHAp particles, and slowed the rate of degradation relative to CA. Despite this, both materials were completely degraded after 6 weeks due to matrix disintegration (Fig. 2A). This process coincided with the period of bone preparation and lamellar bone deposition in the extraction sockets, ensuring that the

rate of material degradation corresponds with the rate of osteogenesis (*Comesaña et al., 2015*).

### *In vitro* cytotoxity

In accordance with the Chinese national standard "Standard Protocol for Evaluation of Cytotoxicity of Materials" (GB/T 16886.5-2003), a medical biological material meets the cytotoxicity requirement if the cell viability is greater than 80%. In this study, the cell viability was greater than 90% at all experimental concentrations (Fig. 2B). The absence of red fluorescence in the cell fluorescence images (Fig. 2C) indicates that the majority of cells were viable. These findings indicated the cytocompatibility of CA and nCA.

### *In vitro* hemocompatibility

The hemolytic activity assay was then used to determine the *in vitro* hemocompatibility of the particles. Using Triton X-100 as a positive control and PBS as a negative control, the hemolysis ratios of the particle samples were evaluated. Compared to the typical bright red color of the Triton X-100 group (Fig. 2D), both the particles and PBS groups exhibited a light-red supernatant. In addition, the hemolysis ratios of all samples were less than 2.0% (Fig. 2E), indicating that the natural Alginate and nano hydroxyapatite particles have excellent hemocompatibility.

### nHAp enhanced CA osteogenic differentiation *in vitro*

Bone tissue engineering materials play a critical role in the regeneration and reconstruction of bone defects. The ideal bone tissue engineering material should not only be biocompatible but should also promote the differentiation of precursor cells into osteoblasts (*Nie et al., 2019*) in order to create a favorable osteogenic microenvironment.

To evaluate the osteogenic induction ability of nCA, ALP (Fig. 3A) and ARS (Fig. 3B) staining were performed *in vitro*. ALP is an early marker of osteogenic differentiation (*Li et al., 2016*), whereas ARS staining is a late marker of mineralization (*Alamán-Díez et al., 2023*). Herein, the nCA group showed the highest staining intensity with the largest proportion of positively stained areas, indicating the enhanced osteogenic differentiation of precursor cells. These findings suggest that nCA can effectively promote the differentiation of BMSCs into osteoblasts, thereby enhancing the mineralization of bone tissue.

### *In vivo* biocompatibility

Biocompatibility is an essential characteristic of biomaterials used in tissue engineering and regenerative medicine. Herein, subcutaneous implantation was used to assess the *in vivo* biocompatibility of granular substances. The H&E staining showed that after 7 days of implantation, subcutaneous tissue had entered the particles, indicating that the granular materials had excellent integration properties. The presence of nHAp in the dark purple areas beneath the skin (Fig. 4B) indicates that nCA is an efficient nHAp carrier. In general, the inflammatory response in the nCA group was comparable to that in the CA group (Fig. 4A), in that both were relatively mild. These results indicate that the addition of nHAp had no effect on the biocompatibility of the substrate material.

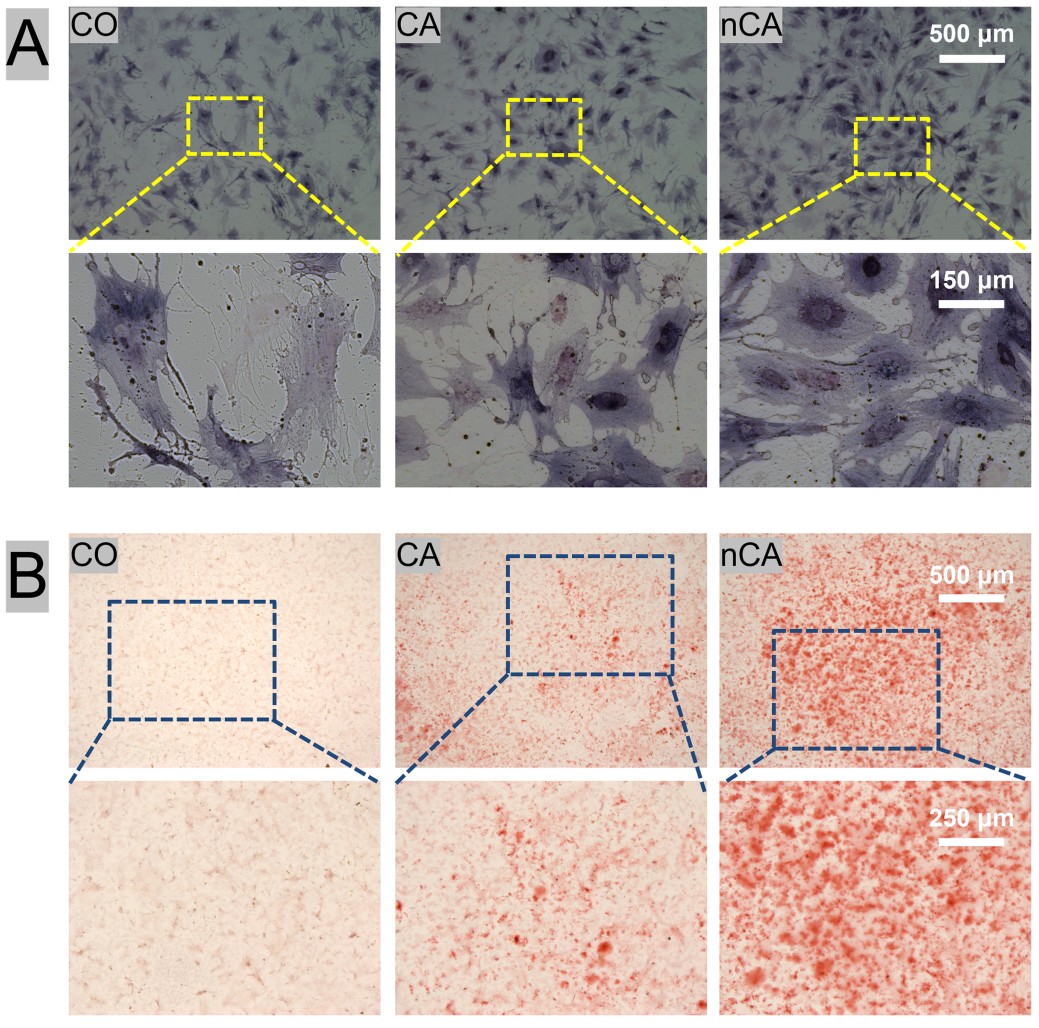

**Figure 3 Osteogenic potential of BMSCs with CA and nCA *in vitro*.** (A) Gross-micrographs of ALP staining. (B) Gross-micrographs of ARS staining.

## *In vivo* hemostatic ability

The tooth extraction process in rats typically begins with a comprehensive examination of the tooth and surrounding tissues to determine the position and condition of the tooth (Fig. 4C), as well as to assess the likelihood of complications during the procedure. Once the rats were fully anesthetized, the gum (Fig. 4D) was separated from the tooth, and the tooth was gently and carefully dislocated from the socket using extraction forceps (Fig. 4E); accordingly, broken teeth are ruled out. The success rate of this experiment is 100%, and no teeth were broken in this study. After the tooth was successfully extracted (Fig. 4F), the extraction site was filled with various particles (Fig. 4G). Finally, the extraction wounds were checked to ensure that they were properly treated and were no longer bleeding (Fig. 4H). Post-extraction bleeding and alveolar ridge resorption are common dental complications, especially in anticoagulant-treated patients (*Morimoto et al., 2016*). Herein,

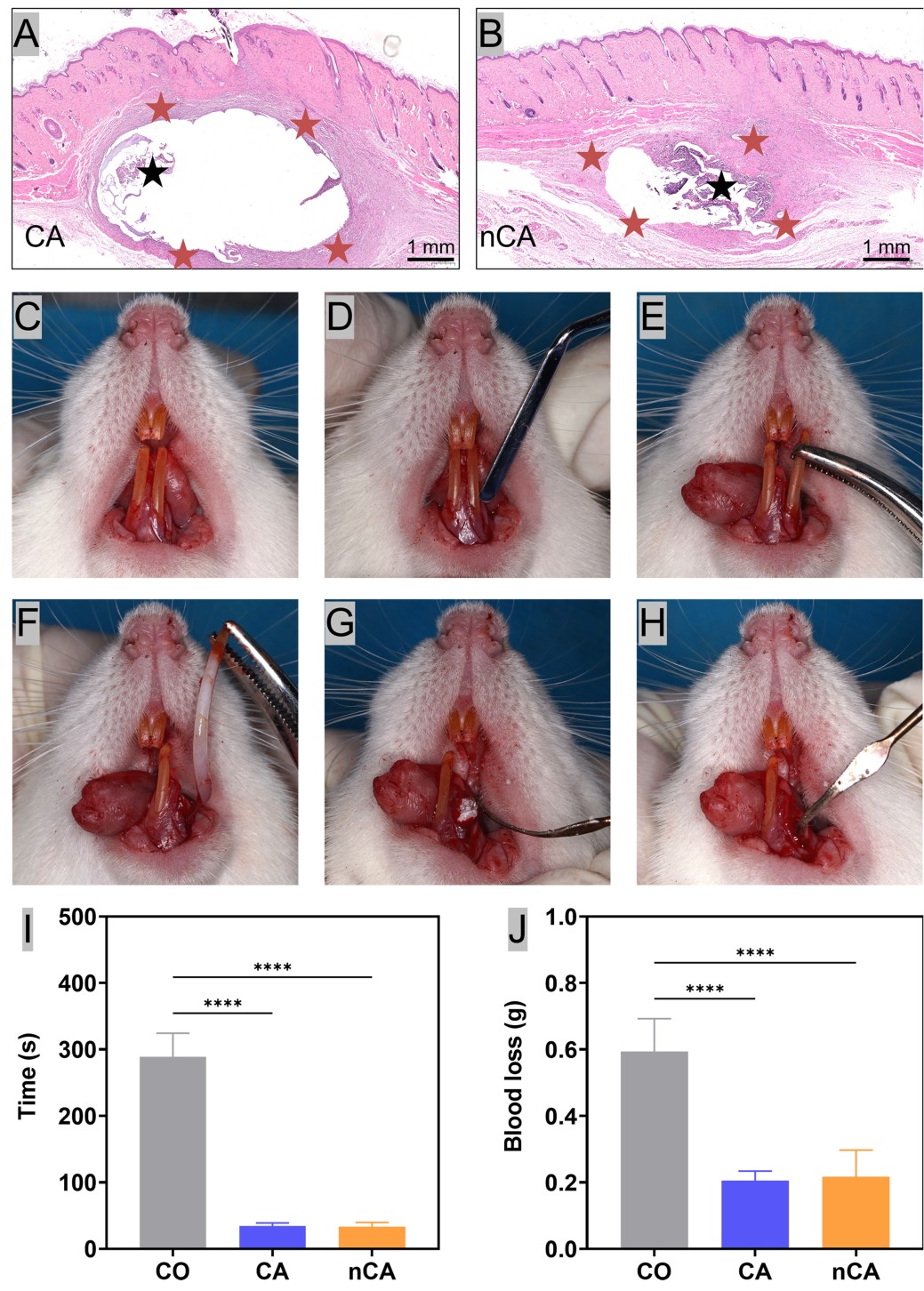

**Figure 4** *In vivo* **biocompatibility and hemostatic ability.** Representative H&E images of CA(A)/nCA (B) particles after 7 days of percutaneous implantation in SD rats. The black and red stars represent the undegraded particle material and the resulting inflammatory cell layer, respectively. (C–H) Procedure of filling materials in extraction sockets. (I) Bleeding time. (J) Amount of blood loss (****$p < 0.0001$).

we compared the hemostatic effects of CA and nCA by measuring blood loss and bleeding time following extraction. The results demonstrated that both CA and nCA possessed excellent hemostatic properties, as evidenced by a reduction in blood loss and bleeding time compared to the CO group, which stopped bleeding with compression using conventional cotton rolls (Figs. 4I and 4J). The increased hemostatic potential of nCA can be attributed to a number of factors. First, the calcium ions in nHAp promote coagulation by activating endogenous and exogenous coagulation pathway factors and converting fibrinogen to fibrin monomers (*der Weijden, Dell'Acqua & Slot, 2009*). In addition, the freeze-dried particles have a tendency to absorb water from the blood, thereby promoting additional coagulation by aggregating red blood cells and platelets. When the nCA granules had transformed into a blood-filled hydrogel, they may have exerted sufficient pressure to stop bleeding.

## Micro-CT evaluation of extraction socket

In this study, the surgical sites were filled with materials after extraction, and the results were evaluated 28 days later. In the CO group, new bone formation was predominant around the alveolar fossa and absent in the central region (Fig. 5A). In contrast, a greater proportion of new bone formed around the alveolar fossa and filled the extraction sockets evenly in the nCA group (Fig. 5C). Similar outcomes were observed in the CA and CO groups (Fig. 5B). In the CO and CA groups, the absence of new bone formation on the external and labial sides of the extracted socket (Figs. 5D and 5E) may be attributed to the absence of protection from filling material, which prevented the formation of a stable blood clot, resulting in impaired osteogenesis (*Cho et al., 2021*). In contrast, the nCA group has superior osteogenesis overall (Fig. 5F) and in the outer area of the socket. In the nCA group, the presence of nHAp promoted bone regeneration.

After 28 days, the nCA group had significantly greater x-ray opacity in the extraction sockets compared to the CO and CA groups. BV/TV and BMD (Figs. 5G and 5H) were also significantly increased in the nCA group. Moreover, the Tb.Sp (Fig. 5I) was 2.2 times higher in the CO group than in the nCA group. The Tb.Th (Fig. 5J) and Tb.N (Fig. 5K) exhibited a higher trend in the CA group and nCA group *vs* the CO group, but no statistical significance was observed.

Bone healing is a complex and multi-factorial coordinated physiological process that is triggered by the formation of a blood clot following injury. The absence of a blood clot would significantly delay the healing process (*Claes, Recknagel & Ignatius, 2012*). Thus, clot aggregation not only serves as an initial fibrin scaffold for supporting cell migration and adhesion but also as a temporary source of bone-regenerating growth factors (*Wang et al., 2017*). With these considerations in mind, we have developed nCA, which consists of calcium alginate and calcium ions in hydroxyapatite, to promote the formation and stabilization of blood clots and, ultimately, bone regeneration. nCA was more resistant to degradation despite the fact that it was initially exposed to the oral environment without being sutured. In addition, both nHAp and CA were found to be closely associated with nCA's capacity for bone regeneration. In previous studies, both nHAp and CA induced osteogenesis in BMSCs and exhibited good bone regeneration properties (*Tal et al., 2008*).

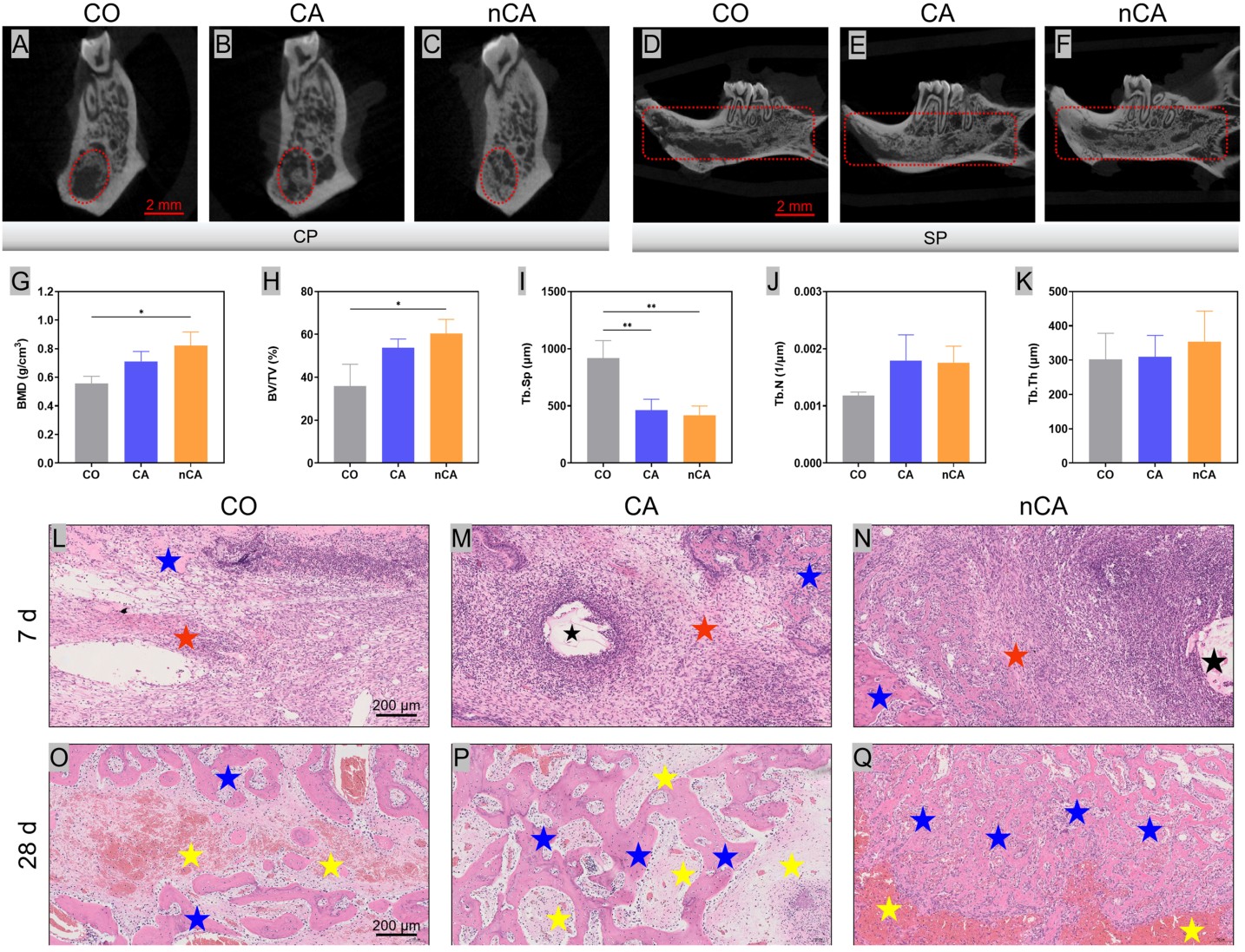

**Figure 5 nCA promoted osteogenesis *in vivo*.** (A–C) Analysis of the lateral coronal section of the extraction. (D–F) Analysis of the sagittal section of the extraction side. (G–K) New bone quantitative data (*$p < 0.05$, **$p < 0.01$). (L–N) H&E staining images of the alveolar ridge in different groups for 7 days. (O–Q) H&E staining images of the alveolar ridge in different groups for 28 days. The black stars represent the undegraded granular material, the red stars the layer of inflammatory cells produced, the yellow stars the fibroblastic tissue, and the blue stars the newly generated bone tissue.

Correspondingly, the migration of stem cells may also be influenced by the growth factors released from the blood clot. Thus, nCA immobilized the blood clot and became a temporary source of growth factor release, as well as an initial structure for cell migration and adhesion (*Kopf et al., 2015*).

## Histological analysis

After 7 days of experimentation, the CO group demonstrated new bone formation at the lingual edge of the extraction socket (Fig. 5L), along with scattered inflammatory cells and multiple cavity areas. In contrast, the CA group (Fig. 5M) displayed undegraded CA particles at the extraction site, infiltration of histiocytes at the margins, and a greater

amount of new bone formation without cavitation throughout the extraction socket. The nCA group also exhibited (Fig. 5N) the presence of undegraded nCA particles in the extraction socket, along with histiocyte infiltration at the margins and an abundance of hydroxyapatite granules. A large number of osteoblasts were present in the nCA group, and the overall osteogenic effect was significantly superior to that of the CO and CA groups. After 28 days, the CO group displayed a significant increase in the quantity of new bone (Fig. 5O), but the CA and nCA groups had significantly more new bone trabeculae. Notably, new bone formation in the CA group (Fig. 5P) tended to fill the defect, with new tissue distributed across the socket. In contrast, bone formation in the CO group was predominantly confined to the edges of the sockets, with the central region filled with blood clots. Additionally, the majority of the defect area was filled with fibrous connective tissue, and new bone formation was significantly slower compared to the CA group. The histological sections of the CA group revealed a greater amount of new bone formation, whereas in the nCA group (Fig. 5Q), the new bone was distributed throughout the extraction socket with fewer blood clots and fibrous tissue. With a more uniform distribution of osteoblasts, the microstructure of the newly formed bone was relatively well-ordered, with a more uniform distribution of osteoblasts. Thus, the nCA particles that were about to be degraded completely could be clearly observed.

No erythrocyte lysis was observed in either the CA or nCA groups, indicating the biocompatibility of both types of materials. The interaction between blood and implanted materials can have a significant effect on the results of implantation. Histological and micro-CT results have further validated the potential of CA and nCA granular materials for both hemostatic and osteogenic functions. In spite of the promising results of the animal model study, additional validation in clinical trials would be required to confirm the efficacy and safety of the composite material. These findings may have important implications for the development of novel biomaterials for bone regeneration and wound healing.

## CONCLUSION

Herein, nano-hydroxyapatite-loaded calcium alginate particles have been developed. The prepared particles have a uniform particle size, can absorb water rapidly, and are easy to preserve. The prepared particles are complexed with nano-hydroxyapatite that has a higher solubility, thereby producing an osteogenic microenvironment with a high calcium ion concentration. The presence of calcium ions activates intrinsic coagulation pathways and platelets and stimulates thrombin production, which significantly improves post-extraction hemostasis. In addition to exhibiting good biocompatibility and osteogenic differentiation *in vitro*, the nCA particles also provided an osteoconductive and osteoinductive microenvironment to promote bone formation *in vivo*. The improved alveolar bone regeneration performance may be associated with the increased osteogenic capacity of bone marrow mesenchymal stem cells, providing a new strategy for clinical hemostasis and bone regeneration after tooth extraction based on the easy availability of this material.

### Funding

This work was supported by the Medical Scientific Research Foundation of Guangdong Province (B2023184 and A2020290), GuangDong Basic and Applied Basic Research Foundation (2019A1515110511). The funders had no role in study design, data collection and analysis, decision to publish, or preparation of the manuscript.

### Grant Disclosures

The following grant information was disclosed by the authors:
Medical Scientific Research Foundation of Guangdong Province: B2023184 and A2020290.
GuangDong Basic and Applied Basic Research Foundation: 2019A1515110511.

### Competing Interests

The authors declare that they have no competing interests.

### Author Contributions

- Gang He conceived and designed the experiments, performed the experiments, analyzed the data, prepared figures and/or tables, authored or reviewed drafts of the article, and approved the final draft.
- Zhihui Chen conceived and designed the experiments, performed the experiments, analyzed the data, prepared figures and/or tables, authored or reviewed drafts of the article, and approved the final draft.
- Luyuan Chen conceived and designed the experiments, performed the experiments, analyzed the data, prepared figures and/or tables, and approved the final draft.
- Huajun Lin performed the experiments, prepared figures and/or tables, and approved the final draft.
- Chengcheng Yu performed the experiments, prepared figures and/or tables, and approved the final draft.
- Tingting Zhao performed the experiments, prepared figures and/or tables, and approved the final draft.
- Zhengwen Luo performed the experiments, prepared figures and/or tables, and approved the final draft.
- Yuan Zhou performed the experiments, prepared figures and/or tables, and approved the final draft.
- Siyang Chen performed the experiments, prepared figures and/or tables, and approved the final draft.
- Tianjiao Yang performed the experiments, prepared figures and/or tables, and approved the final draft.
- Guixian He performed the experiments, prepared figures and/or tables, and approved the final draft.
- Wen Sui analyzed the data, prepared figures and/or tables, and approved the final draft.

- Yonglong Hong analyzed the data, prepared figures and/or tables, and approved the final draft.
- Jianjiang Zhao conceived and designed the experiments, analyzed the data, prepared figures and/or tables, and approved the final draft.

### Animal Ethics

The following information was supplied relating to ethical approvals (*i.e.*, approving body and any reference numbers):

The Southern University of Science and Technology ethical committees approved the study.

### Data Availability

The raw data is available in the Supplemental Files.

The CT data is available at MorphoSource:

https://dx.doi.org/10.17602/M2/M515574

https://dx.doi.org/10.17602/M2/M515612

https://dx.doi.org/10.17602/M2/M515603

https://dx.doi.org/10.17602/M2/M515600

https://dx.doi.org/10.17602/M2/M515584

https://dx.doi.org/10.17602/M2/M515569

https://dx.doi.org/10.17602/M2/M515564

https://dx.doi.org/10.17602/M2/M515559

### Supplemental Information

Supplemental information for this article can be found online at http://dx.doi.org/10.7717/peerj.15606#supplemental-information.

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
