# Peer review of "Hydroxyapatite/calcium alginate composite particles for hemostasis and alveolar bone regeneration in tooth extraction wounds"

_PeerJ, doi:10.7717/peerj.15606_

## Round 0.1 · original submission · Major Revisions

The summary and conclusion of the manuscript need to be rewritten, and the language needs to be polished. Some questions about the graph and experimental design proposed by the reviewer also need to be answered by the authors.

Reviewer 1 ·

Basic reporting

The authors developed infill particles made from hydroxyapatite/calcium alginate. The composite material may have a positive hemostatic effect and promote bone regeneration after tooth extraction. The research topic of this manuscript is of significance for the field of bone tissue engineering. The conceptualized research idea is innovative. The authors provided a sufficient interpretation of their main findings in the discussion section. This manuscript contains some grammatical and spelling mistakes and therefore needs to be proofread and deep edited by a fluent English speaker in the medical field.

Experimental design

The experimental design of the study is rigorous. The research gap was clearly stated in the abstract section. The methodology described in this manuscript is described in sufficient detail.

Validity of the findings

I think the findings are valid and meet the standards of the journal PeerJ. The underlying data are robust and statistically sound. However, the conclusion section is not well stated and thus should be rewritten. The conclusion is suggested to contain a summary of the main findings.

Additional comments

1. In cell cytotoxicity experiments, the form of concentration units in the text and in the figure notes was The concentration unit form should be changed to the conventional form "mg/mL".
2. Why did the authors use a 0.5% concentration of the experimental group in the live/dead staining assay?
3. In the assessment of hemostatic performance, blood loss should be expressed in terms of weight rather than volume.
4. "swelling rate" is mentioned, but the formula is "swelling ratio." Authors should double check it and write it uniformly.
5. I would like to suggest that the abstract be reorganized in a structured way. Several sections should be included in the reorganized abstract: background, objective, methods, results, and conclusion.
6. A clearer scale bar should be provided for Fig 1 B, D, and E.
7. The conclusion of this manuscript is suggested to contain a detailed summary of the main findings.

·

Basic reporting

The writing of this manuscript is clear and understandable. The introduction is written well and emphasizes the research purpose of this manuscript.

Experimental design

This study has a clear theme that should appeal to readers; however, there are several flows of experimental designs needing to be addressed. My specific concerns related to the experimental design are located in the “additional comments”.

Validity of the findings

The findings of this manuscript are confirmed to be valid. However, are there any potential clinical transfer values of these findings for future research? I suggest that the clinical transfer values of these findings should be particularly emphasized in the discussion.

Additional comments

① Figures 1B and E did not match. Figure 1B should be magnified.
② There are only pictures of the hydrogel state of the material when it is not freeze-dried. Images of particles after freeze-drying should be provided.
③ The images of SA and nSA hydrogel states under the microscope should be supplemented.
④ It should be recommended to supplement the identification of each region and the meaning of the representative in the histological staining section.
⑤ The central incisors of SD rats grow throughout their lives. Are they going to grow again after tooth extractions in the experiment? If so, is the experimental result reliable?
⑥ SD rats' central incisors are very easy to break. How did the authors ensure the tooth extraction model's success? Also, what is the success rate?
⑦ The hemolysis rate experiment is recommended to be carried out to demonstrate the hemostatic material's biological safety.
⑧ The methods regarding the ALP and ARS experiments are too simplified and thus need to be expanded.

---

## Round 0.2 · accepted · Accept

The author has addressed the reviewers' concerns.